# Intake of Non-Nutritive Sweeteners in Chilean Children after Enforcement of a New Food Labeling Law that Regulates Added Sugar Content in Processed Foods

**DOI:** 10.3390/nu12061594

**Published:** 2020-05-29

**Authors:** Ximena Martínez, Yazmín Zapata, Victoria Pinto, Camila Cornejo, Martje Elbers, Maaike van der Graaf, Luis Villarroel, María Isabel Hodgson, Attilio Rigotti, Guadalupe Echeverría

**Affiliations:** 1Center for Molecular Nutrition and Chronic Diseases, School of Medicine, Pontificia Universidad Católica de Chile, Av. Libertador Bernardo O’Higgins, Región Metropolitana, 340 Santiago, Chile; xlmartinez@uc.cl (X.M.); yazapata@uc.cl (Y.Z.); vspinto@uc.cl (V.P.); cncornejo@uc.cl (C.C.); martje.elbers@outlook.com (M.E.); maaikevdgraaf@hotmail.nl (M.v.d.G.); arigotti@med.puc.cl (A.R.); 2Hanzehogeschool Groningen, University of Applied Sciences, 9747 AS Groningen, The Netherlands; 3Department of Public Health, School of Medicine, Pontificia Universidad Católica de Chile, Av. Libertador Bernardo O’Higgins 340, Santiago, Región Metropolitana, Chile; lv@med.puc.cl; 4Department of Nutrition, Diabetes and Metabolism, School of Medicine, Pontificia Universidad Católica de Chile, Av. Libertador Bernardo O’Higgins 340, Santiago, Región Metropolitana, Chile; hodgson@med.puc.cl

**Keywords:** non-nutritive sweeteners, intake, schoolchildren, Chile

## Abstract

After enforcement of a new food labeling law in 2016, Chile exhibits a greater offer to reduced sugar products with addition of non-nutritive sweeteners (NNS). Many of these products are consumed by children, who are at greater risk of reaching the acceptable daily intake (ADI) of these food additives. The objective of this study was to evaluate the intake levels of NNS in Chilean schoolchildren after the enactment of the aforementioned law. A total of 250 Chilean children 6–12 years old were surveyed. NNS intake was assessed through a food frequency questionnaire. All children evaluated consumed at least one NNS during the previous month. Sucralose had the highest consumption frequency reaching 99.2%, followed by acesulfame-K (92.8%), stevia (86.0%), and aspartame (85.2%). Aspartame showed the highest median intake, which came mainly from beverages (96%). No children exceeded the ADI of any NNS. Smaller children exhibited a higher body weight-adjusted intake of sucralose, acesulfame-K, stevia, and aspartame (*p* < 0.05). In Chile, a wide range of processed foods with NNSs is available and all schoolchildren evaluated consumed at least one product containing NNS. However, this consumption does not exceed defined ADIs for any of the six sweeteners authorized for food use in Chile.

## 1. Introduction

In the last Chilean National Health Survey performed in 2016–2017, 40% and 31% of people aged more than 15 years were overweight or obese, respectively [1]. The outlook is not very different in Chilean children: based on the nutritional map provided by the National Board of School Assistance and Scholarships in 2019, 15.6% of the first-grade children were obese, reaching a prevalence of 27.9% in fifth graders [2]. These figures are alarming and have led to national implementation of measures to improve the food environment focusing on the quality of food products offered to consumers, with particular emphasis in the child population. 

The interaction between a population and its eating behaviors and food environment has an important role in healthy food preferences [3]. As a consequence, the World Health Organization (WHO) recommends a reduction in sugar (monosaccharide and disaccharide) consumption present in foods below 10% -and a stricter recommendation reaching 5%- of total daily energy intake [4]. This decrease in sugars or caloric sweetener consumption would attenuate the ongoing increase in body weight and its cardiometabolic consequences [5,6]. Furthermore, a high consumption of processed foods and drinks has been associated with obesity and other diet-related non-communicable diseases [7].

In 2012, the Congress of Chile approved Law #20,606 regarding “Nutritional Composition of Foods and their Advertisement” [8], which was enacted in June 2016 in three phases that gradually increased the reduction requirements of critical nutrient content in foods. This law establishes the implementation of obligatory frontal warning labeling on pre-packaged processed foods that exceed pre-established limits for sugars, saturated fats, sodium, and energy content. It also restricts marketing and prohibits school sales of foods that go over these thresholds. The real impact of this normative should be evaluated in the long-term. A recent study showed a favorable trend in reducing the availability of processed food depicting warning seals [9]. This trend may be explained by product reformulation led by the food industry itself, which is moving towards decreased usage of warning labels after the initial enforcement of this law [9]. Furthermore, similar food warning-label regulations have been/will be adopted in other Latin American countries (Peru, Uruguay, Brazil, and Mexico) and Israel.

Facing this evolving regulatory scene in Chile, non-nutritive sweeteners (NNS) have been used as an effective dietary tool in the reformulation of products, providing sweet taste in processed foods, such as beverages, ready-to-eat cereals, and sweets snacks, and avoiding the caloric contribution derived from the addition of sugars. This approach allows to fulfill the established normative and eliminate the “high in sugars” label from the frontal face of these products, many of them directed toward the child population [10].

The impact on health of increased intake of these NNS is still unclear [11]. Additional research is needed to obtain convincing evidence about its efficacy in the long-term as well as its lack of negative effects derived from chronic NNS use in high dosages [12,13,14]. Regarding their role in body weight control, studies with NNS are contradictory: even though there is evidence that replacing sugary drinks with sugar-free and NNS-supplemented drinks reduce significantly the gain of body weight [15,16], other observational studies describe that NNS could be associated with increased body-mass index (BMI) and cardiometabolic risk [17,18]. Therefore, additional interventional clinical trials are needed to confirm or deny their beneficial or detrimental effects on human health.

With regard to the pediatric population beyond industrial food-associated NNS intake, another study analyzed the content of NNS in breast milk in women who did not explicitly inform intake of these additives [19]. The presence of saccharin, sucralose, and acesulfame-K in human milk, therefore, suggests exposure to these NNS in breastfed babies. Within the potential effects of early exposure to sweeteners, the impact over the intestinal microbiome has been reported [14] as well as in the definition of food choices and future dietary patterns since taste preferences are formed in the early years [20]. However, the position of the Academy of Nutrition and Dietetics of USA on the utilization of NNS indicates that they can be used within a healthy diet according to preferences and individual health goals [21].

To assure its safe use, each NNS has to be approved at an international level, through safety assessments performed by the Food and Agriculture Organization (FAO) of the United Nations, the World Health Organization (WHO), and the Joint FAO-WHO Expert Committee Report on Food Additives (JECFA). In Chile, the use of NNS in processed foods is regulated by the Sanitary Regulation for Food Products. This regulation allows the use of saccharin, potassium acesulfame, cyclamate, aspartame, sucralose, stevia, alitame, and neotame, each one of them with its respective admissible daily intake (ADI) expressed in mg/kg of body weight according to FAO/WHO recommendations [22]. The ADI does not actually represent a maximal level of allowed intake, but it indicates a margin of safe daily consumption based on animal studies over a lifetime without having an appreciable health risk. Thus, occasionally surpassing this intake limit level does not necessarily represent a health risk [23,24].

It should be noted that during 2011, before promulgation and enforcement of Law #20,606, NNS intake was assessed in Chile. This study demonstrated that no Chilean children exceeded the ADI for NNS. Acesulfame-K was the sweetener with the closest level of consumption to its ADI, reaching a maximal adequacy of 92.6% of ADI (13.9 mg/kg/day estimated intake versus ADI of 15 mg/kg/day), followed by sucralose that reached an 82.6% of ADI adequacy (12.4 versus 15 mg/kg/day, respectively) [25]. Thus, almost a decade ago, even though the NNS intake was already significant, consumption did not go over allowed levels in Chilean school boys and girls.

Considering these previous findings and the increased availability of food and beverages with NNS in Chile as a result of the new legislation [8,26], the objective of this study was to update the information on non-nutritive sweetener intake in Chilean schoolchildren.

## 2. Materials and Methods 

An observational cross-sectional study was conducted on 250 children aged 6 to 12 years residing in Santiago, the country capital city located within the Metropolitan Region of Chile, together with their parents or tutors by face-to-face interview and anthropometric evaluation in schools. The sampling was of convenience, stratified according to sex (boys and girls), age (6 to 9 years and 10 to 12 years) and type of educational establishment (public and private) to perform sub-analyses according to these pre-defined demographic conditions.

### 2.1. Subjects

The study included children from 6 (1st grade) to 12 (6th grade) years old and their respective parents or tutors residing in Santiago, who voluntarily agreed to participate in the study and signed the informed consent and assent. The exclusion criteria were children with pathological conditions that required diet modification such as diabetes, celiac disease, lactose intolerance, among others; children in nutritional treatment for body weight control; children with food allergies to cow milk protein or food additives; immigrants with less than a year of residence; and siblings of subjects already included in the study. 

### 2.2. Registry of Foods Containing NNS

We identified 398 NNS-containing products available for regular consumption in supermarkets in Santiago. The database was created with information declared by manufacturers in the ingredient labeling printed on the package of national and imported products as required by the Chilean food sanitary code. This amount of NNS-supplemented food products was 226%, 92%, and 176% superior to findings reported in previous Chilean studies performed in 2011, 2013, and 2014, respectively [25,27,28]. These NNS containing products were classified into 5 categories: (a) dairy products, (b) non-alcoholic drinks/beverages, (c) cereals, (d) desserts, and (e) others (sweetened products that were not included in other categories). Overall, combined dairy products (24%), cereals (21%), and non-alcoholic drinks (19%) represented almost two thirds of the registered products that contained NNS (Figure 1). 

The NNS content in each product was obtained from the information declared in the nutritional labeling. Next, the content of NNS informed in the nutritional label was verified with producers, particularly in the case of stevia, whose content must be reported specifically as mg equivalents of steviol to be appropriately compared with defined ADI for this NNS. Nineteen percent of products containing stevia (6.5% of total products) were misreported as steviol glycosides, which was confirmed by food producers, and conversion was made to mg equivalents of steviol as recommended.

### 2.3. Assessment of NNS Intake

The food frequency questionnaire (FFQ) previously reported in 2018 to investigate the consumption of NNS [29] was locally adapted. NNS-FFQ is a brief questionnaire that can be administered among diverse participants at individual and population levels to quantitatively measure routine NNS intake. NNS-FFQ was implemented through a web page including photos of each product with NNS included in the registry to facilitate product identification by children and their parents/tutors. The application of this instrument was done by trained nutritionists through face-to-face interviews with parents/tutors and children surveyed in this study.

### 2.4. Anthropometric Evaluation

In children, body weight was measured using an electronic scale calibrated to 200 kg with an accuracy of 0.1 kg and the height was measured with a portable stadiometer of 205 cm with an accuracy of 0.1 cm. Weighing was performed with pants or skirt and shirt, discounting 400 g due to clothing, and height was measured during inspiration by lightly touching the top of the head. The nutritional status was evaluated using z-score of body mass index (BMI) according to WHO and Chilean Ministry of Health guidelines for children and adolescents [30]. 

### 2.5. Ethical Approval

The study protocol (ID 18709006) was evaluated and approved by the Research Ethics Committee of the School of Medicine at Pontificia Universidad Católica de Chile (“Comité Ético Científico de la Facultad de Medicina de la Pontificia Universidad Católica de Chile”) on 7 August 2018.

### 2.6. Statistical Analysis

All the data of NNS intake were corrected by body weight. The normality of the NNS intake data was analyzed by Kolmogorov–Smirnov test. Since every NNS intake showed a non-normal distribution, non-parametric comparison were used, such as Chi-squared (for categorical variables) and Mann-Whitney U (for continuous variables) tests. Multivariate linear regression analyses were conducted to evaluate the association between NNS intake and sociodemographic variables considered in this study. All statistical analysis was performed with the SPSS® Statistics software, version 24 (IBM Corporation, Armonk, NY, USA). A comparison difference was considered statistically significant when the *p* value was less than 0.05.

## 3. Results

The study included 250 children from first to sixth year of primary education, recruited from 8 public and private schools in Santiago at the Metropolitan Region in Chile. The mean age of the evaluated children was 9.1 ± 1.8 years-old, with a greater representation of boys (54%) than girls as well as children studying in public schools (66%) (Table 1).

**When assessing nutritional status, 38.8**% of the total sample of children had overweight or was obese (Table 1), with no differences in the distribution of nutritional status according to sex or age groups. However, when comparing types of educational establishment, important differences were found: 16% of children attending private schools exhibited overweight but none was obese, while 28% and 24% of schoolchildren studying in public schools had overweight and obesity, respectively (*p* < 0.001).

### 3.1. Overall NNS Intake 

All schoolchildren (100%) reported consumption of food or beverages that contained some type of NNS in their formulation as a sugar substitute. The NNS with the highest consumption frequency was sucralose (99.2%), followed by acesulfame-K (92.8%), stevia (86.0%), and aspartame (85.2%). Cyclamate and saccharin had considerably lower levels of intake, with consumption reported in 12.0% and 10.8% of children, respectively.

Considering that each of these NNS have a different ADI by body weight, Table 2 shows daily intake of each NNS adjusted by body weight. All the children included in this study reported safe NNS intakes compared to the defined ADIs. Sucralose and stevia were the NNS that presented the highest levels of intake with respect to their ADIa, but always at admissible levels, with maximal intakes of 62% and 60% in relation to their corresponding ADIs (Table 2).

### 3.2. NNS Intake According to Sociodemographic Variables and Their Main Food Sources

#### 3.2.1. Sucralose

Sucralose intake was not associated with differences in sex, type of educational system, or nutritional status. Only significant differences were found by age, where younger children, aged 6 to 9 years old, consumed more sucralose than older children (1.7 vs. 1.0 mg/kg·day, respectively, Table 3). Also, the adequacy to ADI of sucralose analyzed in an adjusted model, which included sociodemographic variables of the study and nutritional status, maintained a significant association with age, where children aged 6 to 9 years old exhibited higher adequacy to ADI (Table 4). Sucralose intake came mainly from beverages (48%) and dairy products (39%) (Figure 2).

#### 3.2.2. Acesulfame-K

Acesulfame-K intake in this study only presented significant differences according to age groups, where younger children had higher consumption than older children (1.0 vs. 0.7 mg/kg·day, respectively, Table 3). This association remained significant when analyzing the adequacy to ADI of acesulfame-K in the adjusted model that included sociodemographic variables and nutritional status (Table 4). Beverages provided 94% of the intake of acesulfame-K (Figure 2).

#### 3.2.3. Stevia

Stevia intake showed significant differences by type of educational establishment and by age. Children attending private schools had a 50% higher body weight-adjusted intake than children enrolled in public schools (Table 3). In addition, younger children presented almost three times the consumption (adjusted for weight) than the older ones (0.31 vs. 0.13 mg/kg·day, respectively, Table 3). The association between the adequacy of stevia to its ADI and the study variables in the adjusted model remained significant with regard to age, but it was lost by type of educational system (Table 4). With respect to food sources of stevia, it mainly came from dairy products (43%) and beverages (36%) (Figure 2).

#### 3.2.4. Aspartame

Aspartame intake only showed significant differences according to age, where older children reported lower consumption (adjusted for weight) than younger children. No differences were observed by sex, type of educational establishment, or nutritional status (Table 3). The adequacy of aspartame to its ADI showed a significant association with the type of school system in the adjusted model: children from private schools consumed less aspartame (adjusted for body weight) than children from public schools (Table 4). Finally, almost all aspartame intake (96%) came from beverages (Figure 2).

#### 3.2.5. Cyclamate and Saccharin

Cyclamate and saccharin had very low frequency of consumption (12.0 and 10.8%, respectively) among Chilean children, and no significant differences were observed based on any of the sociodemographic variables included in this study (Table 3). Analysis of all the study variables included in the adjusted model showed that children from private schools (*p* = 0.027) and children who had overweight or obesity (*p* = 0.015) reported lower adequacy values of cyclamate to its ADI (Table 4). Finally, the intake of these two NNS derived from beverages (61% for cyclamate and 37% for saccharin) and desserts (39% and 63% for cyclamate and saccharin, respectively, Figure 2).

### 3.3. Combined NNS Intake

As indicated in the previous section, all children evaluated were exposed to the consumption of at least one food product containing some type of NNSs. However, the reported consumption levels were safe for all NNSs according to predefined individual ADIs, with maximal levels of consumption that did not exceed 62% of the ADI for any of the sweeteners analyzed.

As a first approach to estimating the total combined exposure to this type of food additives, Figure 3 shows the total intake of NNSs expressed as the sum of percentages of ADI adequacy of each sweetener (or equivalent combined intake of NNSs) consumed by each child. Only 5 children (2% of the sample) in our study reported a combined total consumption of NNSs that exceeded 100% of the sum of the ADIs of each individual NNS.

Finally, a multivariate analysis of the equivalent combined consumption of NNSs with nutritional status and sociodemographic variables showed that children from public schools, of younger age (6 to 9 years old), and without obesity exhibited higher total consumption of sweeteners adjusted by body weight (Table 5).

## 4. Discussion

Public health policymaking is introducing worldwide progressive and sustainable reduction in energy and non-healthy nutrient intake. Chilean Law #20,606 on food labeling fulfilled the third stage of its implementation in June 2019, with the objective of reducing -through frontal warning labels- purchase and consumption of food products high in critical ingredients (saturated fats, sugars, and sodium) and energy content. In particular, the regulation regarding total sugars allows a maximal content of 5 or 10 g of this nutrient per 100 ml of beverages or 100 g of foods, respectively, to avoid the use of warning seals. As a consequence, the Chilean food industry has reformulated many of its products, replacing sugar with additives such as NNS to provide sweet flavor without added calories. Thus, there is concern that our population, particularly children, may be overexposed to NNS reaching intake levels above adequate daily limits. 

This study reports a large, but admissible, exposure to sugar substitutes among schoolchildren living in Santiago, the country capital city located within the Metropolitan Region of Chile. All children surveyed consumed some type of NNS provided by beverages, dairy products, or other processed foods. Despite this widespread consumption, individual NNS intake levels were beneath each of the pre-established specific ADIs.

The frequency and amount of consumption of NNSs vary according to the availability of new foods on the market that contain these ingredients. In our study, sucralose was the sweetener with the highest consumption frequency likely as a consequence of high exposure due to its most frequent use in the food industry: 75% of the 356 products found with NNS in the market had sucralose in their formulation. 

With regard to the sources that provided NNSs during food intake, dairy products and beverages were the major contributors to the intake of sucralose and stevia, even though these NNSs were also present in some types of cereals and other sweet snacks. Drinks/beverages mainly provided aspartame and acesulfame-K, both with the highest absolute median intakes, and aspartame being the NNS with the maximal median consumption. On the other hand, it draws attention the low intake of saccharin and cyclamate (only 6% of our study sample), which was derived mainly from the consumption of very low cost beverages and some packaged desserts. 

Despite high availability and exposure to NNSs, all school children included in this study reported appropriate intake of each individual type of these food additives compared to predefined ADIs. Even those showing the highest levels of intake (i.e., sucralose and stevia) exhibited consumption at admissible levels reaching less than two thirds of their corresponding ADIs. However, 5 children (2%) in our study reported a combined total consumption of NNSs that exceeded 100% of the sum of the ADI of each individual NNS. The possible additive or synergistic impact originated by consumption of various NNSs is unknown, nor does if exceeding 100% of the sum of the individual ADIs has any physiological or disease risk implications. 

As shown by Wilson et al. in 2019 [31], parents are unlikely to even know what NNS are. Thus, they are mostly unaware of the presence of NNS in foods and do not regulate the amount of intake by their kids. This is even more important in Chile because products containing NNS avoid the “High in sugars” warning seal, which in turn increases sales and consumption of these food additives.

On the other hand, a greater exposure to products reformulated with NNSs may be modeling child population towards a greater taste for sweet foods [32], since these additives improve palatability of many preparations by enhancing sweet taste. However, more research is required to analyze whether consumption of NNSs can indeed modulate brain responses leading to a greater predisposition to sweet flavor and consumption of sweet products [33]. 

In this study sample, we did not find any association between NNS intake adjusted by body weight and nutritional status of children. Several studies indicate that the almost null caloric supply of NNSs could favor an energy intake compensation due to a higher food intake [34]. In contrast, some studies indicate that the use of NNSs does not modify hunger and satiety levels, and such energy compensation is partial, maintaining low food energy supply compared to the addition of sucrose [35,36,37]. This unresolved controversial issue needs further evaluation in clinical trials to update current guidelines [18].

To assess the level of sweetener intake worldwide, Martyn and collaborators [38] conducted a review providing a broad picture of consumption in different world populations. While cyclamate, acesulfame-K, steviol glycosides, and saccharin exceed safety recommendations in some populations, it occurred mostly among high-level consumers and/or specific sub-groups of the population [38]. In Asia, the consumption of sweeteners is below recommendations as related to ADIs but a high intake of cyclamate (95% of its ADI) was detected in Chinese adolescents due to consumption of canned fruits formulated with this sweetener. However, most of the studies have been conducted in Europe and North America, where the intake is usually low and remains within safe ranges. The groups of greatest concern are young children with phenylketonuria or consumers of special feeding formulas since they largely exceed the ADI for acesulfame-K. 

In Latin America, some studies have been carried out on this subject [38] even though assessments were conducted in small samples that were not nationally representative. Despite of overall intake of NNS under the ADIs, some consumption exceeding admissible levels was found for cyclamate and saccharin in children and diabetic individuals, respectively [38]. 

Two previous studies in Chilean schoolchildren with age ranges similar to our work have been reported [25,27]. In comparison with the previous study conducted in a similar population [25], our work found a reduction in the median intake adjusted for body weight in all sweeteners evaluated as well as lower levels of ADI adequacy, with the exception of saccharin (6% in the previous report versus 13.6% of adequacy relative to ADI for saccharin in our study). These differences could be attributed to changes in the formulation and implementation of new sweetener mixtures by the food industry in recent years. On the other hand, Durán (2011) [25] and Hamilton (2013) [27] did not detect stevia consumption in their surveys, while our study indicates that stevia currently ranks third in frequency of consumption. This difference may be related to a more recent shift towards greater offer of this NNS by the industry and a better perception of consumers for this sweetener due to its natural origin, which may positively influence purchase preferences. As a consequence of more restrictive regulations and measures against sugar consumption, as occurred in Chile, it is very likely that the availability, exposure, and consumption of NNSs will continue to increase in different regions of the world.

The use of information regarding the actual content of sweeteners reported in the products included in our registry combined with their classification by food groups, is a strength of this study. This approach made possible to develop a face-to-face interview using an online platform with photos and recognizable brands, facilitating the identification of specific products, minimizing the interviewer’s interpretation of data provided by participants, and allowing a more realistic estimate of the intake of NNSs.

One weakness of this study derives from the use of a sample that is not representative at regional or national population levels. Even so, our 250 children sample allow us to state with 95% confidence that the probability that a child from 6 to 12 years of age living in Santiago of Chile exceeds the ADIs defined for any of the NNS is less than 0.1%. On the other hand, the consumption frequency survey was specifically focused on the intake of foods containing non-nutritive sweeteners, thus it was not possible to compare consumption preferences between foods with and without sugar or establish differences between consumers and non-consumers of sweeteners. In addition, no information was obtained on other sugar substitutes such as polyols (e.g., mannitol, sorbitol, and xylitol, among others), which are not included in the current nutritional labeling of foods. 

It remains important to keep monitoring NNS intake based on evolving regulations to reduce the levels of sugar consumption. This surveillance should be particularly relevant for high-risk individuals, such as diabetic children, pregnant women, diabetic adults and other subjects with specific dietary requirements to timely ensure risk management. 

## 5. Conclusions

This study reports a large, but admissible, exposure to sugar substitutes among schoolchildren living in the city of Santiago within the Metropolitan Region of Chile. All children surveyed consumed some type of NNS provided by beverages, dairy products, or other processed foods. Despite this widespread consumption, individual NNS intake levels were beneath each of the pre-established specific ADIs. More research is needed to assess the possible additive or synergistic impact of the long-term consumption of various NNSs.

## Figures and Tables

**Figure 1 nutrients-12-01594-f001:**
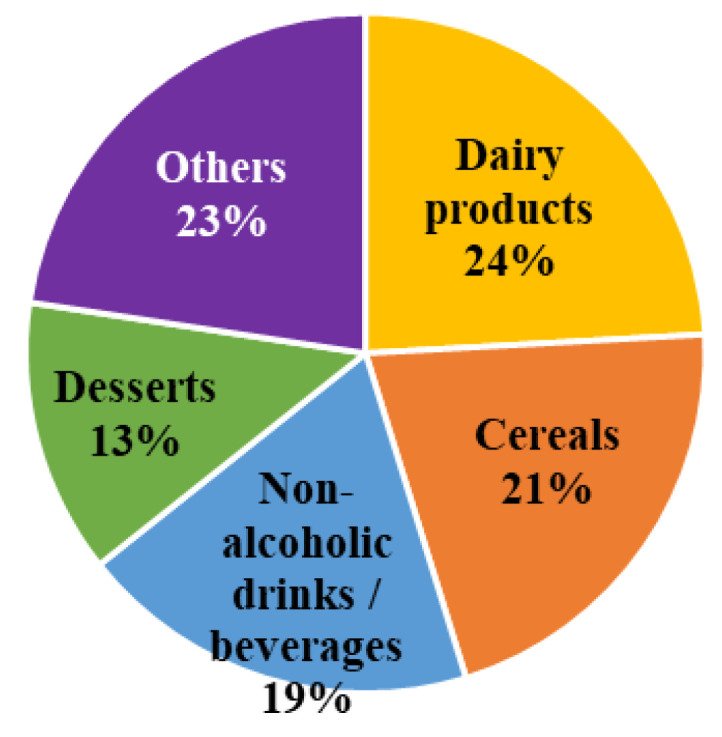
Main food group classification of 398 products containing non-nutritive sweeteners available in supermarkets located in the city of Santiago within the Metropolitan Region of Chile.

**Figure 2 nutrients-12-01594-f002:**
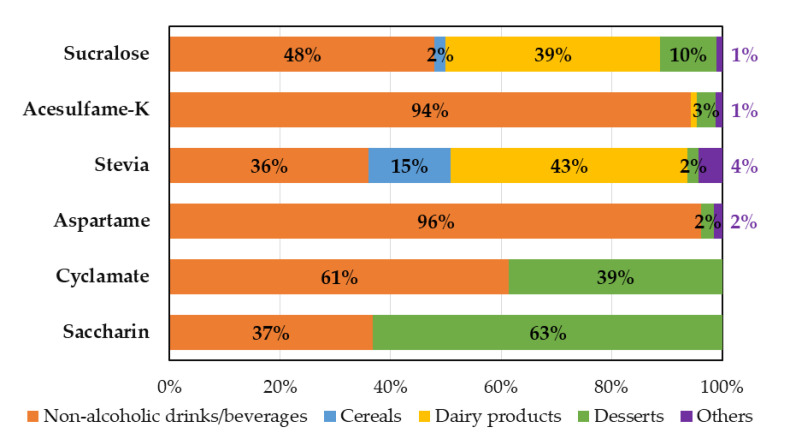
Food sources of non nutritive sweetener intake in Chilean schoolchildren aged 6 to 12 years living in the city of Santiago within the Metropolitan Region of Chile.

**Figure 3 nutrients-12-01594-f003:**
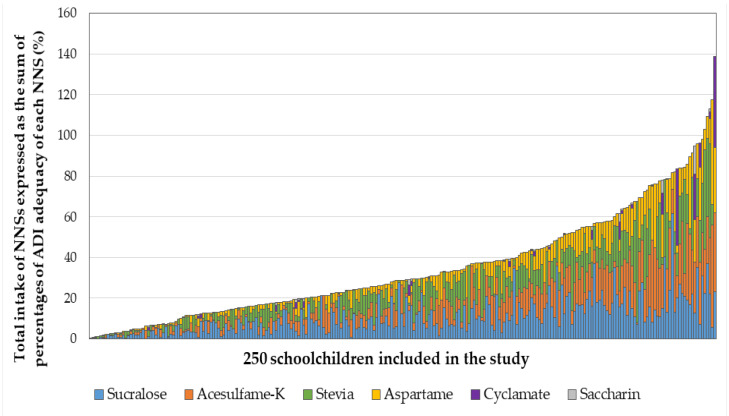
Combined total exposure to non nutritive sweeteners in 250 Chilean schoolchildren aged 6 to 12 years living in the city of Santiago within the Metropolitan Region of Chile. Schoolchildren are presented in order from lower to highest total intake of NNS.

**Table 1 nutrients-12-01594-t001:** Sample description of 250 schoolchildren aged 6 to 12 years living in the city of Santiago within the Metropolitan Region of Chile.

Parameter	Number of Children (*n*)	Frequency (%)
Sex		
Girls	116	46.4
Boys	134	53.6
Type of school system		
Private	90	36.0
Public	160	64.0
Age		
6 to 9 years old	147	58.8
10 to 12 years old	103	41.2
Nutritional status		
Underweight	26	10.4
Normal weight	127	50.8
Overweight	59	23.6
Obese	38	15.2

**Table 2 nutrients-12-01594-t002:** Frequency, median, and maximal intake adjusted by body weight of each non-nutritive sweeteners in Chilean schoolchildren living in Santiago of Chile.

NNS	Number (%) of Products Containing each NNS	Intake Frequency (%)	Median (Interquartile Range) Intake (mg/kg·Day)	Maximal Intake (mg/kg·Day)	ADI (mg/kg·Day)	Maximal Adequacy to ADI (%)
Sucralose	291 (73%)	99.2	1.32 (0.67–2.32)	9.24	15.0	61.63
Acesulfame-K	70 (18%)	92.8	0.88 (0.25–2.10)	7.60	15.0	50.69
Stevia (mg steviol eq.)	139 (35%)	86.0	0.24 (0.04–0.47)	2.39	4.0	59.64
Aspartame	51 (13%)	85.2	1.42 (0.26–3.55)	20.62	40.0	51.55
Cyclamate	7 (2%)	12.0	0.00 (0.00–0.00)	3.13	7.0	44.75
Saccharin	7 (2%)	10.8	0.00 (0.00–0.00)	0.68	5.0	13.62

**Table 3 nutrients-12-01594-t003:** Non nutritive sweetener intake by sociodemographic variables in 250 schoolchildren aged 6 to 12 years living in the city of Santiago within the Metropolitan Region of Chile.

NNS Intake (mg/kg·Day) Median (Interquartile Range)	Sucralose	Acesulfame-K	Stevia (mg Steviol eq.)	Aspartame
By sex:				
Girls	1.16 (0.66–2.17)	0.88 (0.40–2.17)	0.28 (0.04–0.54)	1.43 (0.44–3.70)
Boys	1.46 (0.68–2.38)	0.88 (0.17–2.04)	0.21 (0.04–0.42)	1.28 (0.12–3.22)
*p*-value	0.423	0.226	0.190	0.134
By type of educational system:				
Private	1.25 (0.55–2.20)	0.72 (0.22–1.92)	0.30 (0.07–0.58)	1.14 (0.24–3.17)
Public	1.36 (0.80–2.38)	0.99 (0.28–2.18)	0.20 (0.04–0.39)	1.56 (0.26–4.20)
*p*-value	0.285	0.313	**0.029**	0.177
By age group:				
6-9 years old	1.69 (0.89–2.38)	0.98 (0.33–2.34)	0.31 (0.10–0.57)	1.74 (0.34–4.41)
10-12 years old	0.97 (0.35–1.66)	0.72 (0.17–1.60)	0.13 (0.02–0.31)	0.93 (0.23–2.51)
*p*-value	**<0.001**	**0.036**	**<0.001**	**0.012**
By nutritional status:				
Under and normal weight	1.37 (0.71–2.36)	0.72 (0.17–2.07)	0.28 (0.04–0.49)	1.30 (0.18–3.39)
Overweight and obesity	1.20 (0.61–2.31)	1.07 (0.40–2.11)	0.19 (0.04–0.39)	1.61 (0.41–3.63)
*p*-value	0.380	0.211	0.250	0.221

Univariate analysis. Non-parametric Kruskal–Wallis test was used to evaluate significant differences associated with demographics and nutritional status. Bold *p* values indicate a statistically significant difference between subgroups.

**Table 4 nutrients-12-01594-t004:** Adjusted association between adequacy to the acceptable daily intake for consumption of each non nutritional sweetener with nutritional status and sociodemographic variables in Chilean schoolchildren.

Sociodemographic Variables		ß	*p*-Value
Adequacy of sucralose consumption to ADI (%)			
Sex	Girls vs. Boys	−0.826	0.439
Type of educational system	Private vs. Public	−1.390	0.241
Age (years)	6-9 vs. 10-12	**5.676**	**0.000**
Nutritional status	Under and normal weight vs. Overweight and obesity	1.602	0.172
Adequacy of acesulfame-K consumption to ADI (%)			
Sex	Girls vs. Boys	0.986	0.386
Type of educational system	Private vs. Public	−1.786	0.075
Age (years)	6–9 vs. 10–12	**2.525**	**0.030**
Nutritional status	Under and normal weight vs. Overweight and obesity	0.704	0.572
Adequacy of stevia consumption to ADI (%)			
Sex	Girls vs. Boys	1.268	0.287
Type of educational system	Private vs. Public	2.042	0.123
Age (years)	6–9 vs. 10–12	**4.898**	**0.000**
Nutritional status	Under and normal weight vs. Overweight and obesity	0.660	0.613
Adequacy of aspartame consumption to ADI (%)			
Sex	Girls vs. Boys	0.741	0.410
Type of educational system	Private vs. Public	**−2.670**	**0.008**
Age (years)	6–9 vs. 10–12	**2.235**	**0.015**
Nutritional status	Under and normal weight vs. Overweight and obesity	0.466	0.637
Adequacy of cyclamate consumption to ADI (%)			
Sex	Girls vs. Boys	−0.039	0.940
Type of educational system	Private vs. Public	**−1.291**	**0.027**
Age (years)	6–9 vs. 10–12	−0.265	0.617
Nutritional status	Under and normal weight vs. Overweight and obesity	**1.403**	**0.015**
Adequacy of saccharin consumption to ADI (%)			
Sex	Girls vs. Boys	−0.116	0.373
Type of educational system	Private vs. Public	−0.206	0.153
Age (years)	6–9 vs. 10–12	−0.138	0.296
Nutritional status	Under and normal weight vs. Overweight and obesity	0.259	0.069

Multivariate analysis adjusted models by sex, age group, type of educational system, and nutritional status. Bold *p* values indicate a statistically significant association between variables.

**Table 5 nutrients-12-01594-t005:** Adjusted association between combined non nutritive sweetener intake (expressed as sum of percentages of ADI adequacy for each NNS) with nutritional status and sociodemographic variables in 250 Chilean schoolchildren living in the city of Santiago within the Metropolitan Region of Chile.

Variables	ß (95% Confidence Interval)	*p*-Value
Sex		
Girls	21% (−41% a 84%)	0.500
Boys	Reference	
Type of educational system		
Private	−7% (−14% **a 0**%)	**0.049**
Public	Reference	
Age		
6 to 9 years old	**15% (8% a 21%)**	**<0.001**
10 to 12 years old	Reference	
Nutritional status		
Under and normal weight	**13% (3% a 22%)**	**0.010**
Overweight	**12% (1% a 22%)**	**0.027**
Obesity	Reference	

Multivariate analysis, linear regression model adjusted by sex, age, type of educational establishment and nutritional status. Bold type *p*-values indicate a statistically significant association between variables.

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
