# Peer review of "Intake of Non-Nutritive Sweeteners in Chilean Children after Enforcement of a New Food Labeling Law that Regulates Added Sugar Content in Processed Foods"

_nutrients, 2020, doi:10.3390/nu12061594_

Round 1

Reviewer 1 Report

Ximena Martínez1 , Yazmín Zapata1 , Victoria Pinto

Intake of non-nutritive sweeteners in Chilean….

Abstract:  Nice reflection of content of paper

Introduction: Good context with regards to the problem the authors seek to address.  I think it will be a lot better presented when the writing quality is improved. Improving our understanding of changes in NNS use in in a population with childhood obesity is a very important task indeed.

42: …between a population ,,,and its….

44: monosaccharide and saccharide…not polysaccharide???   Singular best?

51-54: huge sentence…break up to two?

56: rough wording

74: rough wording

77:exposition-rough wording

91 and 94: remove –

101-103: rough

Methods:

Generally well written, and contains very nice details regarding how the study was performed.  The writing quality could be a bit better.

112: all metropolitan areas of larger cities or just one metro area? You said Santiago later…perhaps include this earlier?

126: Content based on ingredient rank order or is mg content required on food packaging?  Perhaps include a food ingredient label as an example?

Are manufacturers required to share content information? In the US folks are typically not excited to share this information.  Also are these mostly domestically produced food products?  Did international food manufacturers share their content information freely and openly?

Results:

Some great information, though I would probably switch the order of presentation a bit.

Table 2 would be improved in the standard deviation for each NNS could accompany the interquartile range….you also may wish to include the number of products identified that contained each NNS, some products may also have a blend of more than one NNS.

192: Maybe it is just me and my preference for writing these sections would be different.  I would write it like this….

Sucralose intake was not associated with differences in sex, type of educational system, or nutritional status. 

Same for other NNS in results

I like Figure 2 very much and might put that earlier in the results because it helps one understand how they are utilized.  Also I think Cyclamate is spelled incorrectly in the figure.

Figure 3 is compelling, you may wish to enlarge the Y-axis text size or shorted the length…its hard to read it.  Also….on X-axis….I think you are attempting to say that this represents NNS intake for each of the 250 subjects analyzed …no need for (n) in my opinion.

Manuscript is well worth publishing just for the content in Figures 2 and 3!  Bravo!

Discussion: Good general discussion of problem with respect to their findings and those of others.

Line 268 is rough: Public health policymaking is introducing worldwide progressive and sustainable reduction in 269 national intake of energy and some non-healthy nutrients.

I do not recall your previously describing the three stages of this Chilean law, it might be helpful to just say stages 1, 2 and 3 in your introduction so the reader better understands this health policy and its success at weight control.

Line 308: You may also wish to mention that parents are unlikely to even know what NNS are, this is important perhaps with regards to selecting foods that reduce sweet taste exposure.  Parents cannot regulate what their kids eat if they have no idea what NNS are or if they cannot identify them in their foods, if they cannot we may all be eating ever sweeter tasting foods thanks to NNS, great for NNS use and sales, not sure if this is really good for our health right?  Wilson T, Murray B, Price T, Atherton D, Hooks T.  Non-Nutritive (Artificial) Sweetener Knowledge among University Students. Nutrients. 2019; 11(9). pii: E2201. doi: 10.3390/nu11092201

Author Response

RESPONSE TO REVIEWER 1

Introduction:

Good context with regards to the problem the authors seek to address.  I think it will be a lot better presented when the writing quality is improved. Improving our understanding of changes in NNS use in in a population with childhood obesity is a very important task indeed.

42: …between a population ,,,and its….

Corrected in the manuscript: “The interaction between a population and its eating behaviors and food environment has an important role in healthy food preferences”

44: monosaccharide and saccharide…not polysaccharide???   Singular best?

Corrected in the manuscript: “As a consequence, the World Health Organization (WHO) recommends a reduction in sugar (monosaccharide and disaccharide) consumption present in foods down to 10% and a stricter recommendation reaching 5% of total daily energy intake”

51-54: huge sentence…break up to two?

The sentence was broke up into two separated phrases: “This law establishes the implementation of obligatory frontal warning labeling on pre-packaged processed foods that exceed pre-established limits for sugars, saturated fats, sodium, and energy content. It also restricts marketing and prohibits school sales of foods that exceed these thresholds.”

56: rough wording

The English style was corrected in the manuscript: “A recent study showed a favorable trend in reducing the availability of processed food depicting warning seals. This reduction may be explained by product reformulation led by the food industry itself, which is moving to fewer use of warning labels after first enforcement of this law.”

74: rough wording

The English style was corrected in the manuscript: “Therefore, more interventional clinical trials are needed to confirm or deny their beneficial or detrimental effects on human health.”

77:exposition-rough wording

The English style was corrected in the manuscript: “The presence of saccharin, sucralose, and acesulfame-K in human milk therefore suggests exposure to these NNS in breastfed babies.”

91 and 94: remove –

The “-“ symbols were eliminated as suggested: “The ADI does not actually represent a maximum level of allowed intake, but it represents a margin of safe consumption based on animal studies on a daily basis over a lifetime without having an appreciable health risk.”

101-103: rough

The English was corrected in the manuscript: “Considering these previous findings and the increased availability of food and beverages with NNS in Chile as a result of the new legislation [26], the objective of this study was to update the information on non-nutritive sweetener intake in Chilean schoolchildren.”

Methods:

Generally well written, and contains very nice details regarding how the study was performed.  The writing quality could be a bit better.

112: all metropolitan areas of larger cities or just one metro area? You said Santiago later…perhaps include this earlier?

Chile is divided into 16 administrative regions, one of them is called Santiago Metropolitan Region. Within this region, Santiago is the capital city of the country. The study was performed in Santiago, Metropolitan Region.

We fixed the manuscript to make it clear we refer to Santiago as “the capital city of the country located within the Santiago Metropolitan Region of Chile”.

126: Content based on ingredient rank order or is mg content required on food packaging?  Perhaps include a food ingredient label as an example? Are manufacturers required to share content information? In the US folks are typically not excited to share this information.  Also are these mostly domestically produced food products?  Did international food manufacturers share their content information freely and openly?

In Chile, manufacturers are required to indicate the NNS content of foods in the nutritional Information label located at the back of the package. National and international manufacturers have to include this information to sell their products in Chile.

The ranking (Figure 1) does not refer to quantity of NNS contained in each product. It indicates the number of products in each category that contain NNS.

Results:

Some great information, though I would probably switch the order of presentation a bit.

Table 2 would be improved in the standard deviation for each NNS could accompany the interquartile range….you also may wish to include the number of products identified that contained each NNS, some products may also have a blend of more than one NNS.

We used median and interquartile range, instead of mean and standard deviation, because NNS intake did not have a normal distribution.

We added a column with the number as well as % of products containing each NNS. One product can be formulated with more than one NNS.

192: Maybe it is just me and my preference for writing these sections would be different.  I would write it like this…. Sucralose intake was not associated with differences in sex, type of educational system, or nutritional status. Same for other NNS in results

Corrected in the text as suggested.

I like Figure 2 very much and might put that earlier in the results because it helps one understand how they are utilized.  Also I think Cyclamate is spelled incorrectly in the figure.

Figure 2 is shown earlier in the manuscript. The spelling of cyclamate was corrected in the figure.

Figure 3 is compelling, you may wish to enlarge the Y-axis text size or shorted the length…its hard to read it.  Also….on X-axis….I think you are attempting to say that this represents NNS intake for each of the 250 subjects analyzed …no need for (n) in my opinion.

Corrected as suggested.

Manuscript is well worth publishing just for the content in Figures 2 and 3!  Bravo!

Thank you!

Discussion:

Good general discussion of problem with respect to their findings and those of others.

Line 268 is rough: Public health policymaking is introducing worldwide progressive and sustainable reduction in 269 national intake of energy and some non-healthy nutrients.

Replaced with: “Public health policymaking is introducing worldwide progressive and sustainable reduction in energy and some non-healthy nutrient intake”.

I do not recall your previously describing the three stages of this Chilean law, it might be helpful to just say stages 1, 2 and 3 in your introduction so the reader better understands this health policy and its success at weight control.

We have mentioned the 3 stages of the Chilean law in the Introduction: lines 51-52.

Line 308: You may also wish to mention that parents are unlikely to even know what NNS are, this is important perhaps with regards to selecting foods that reduce sweet taste exposure.  Parents cannot regulate what their kids eat if they have no idea what NNS are or if they cannot identify them in their foods, if they cannot we may all be eating ever sweeter tasting foods thanks to NNS, great for NNS use and sales, not sure if this is really good for our health right?  Wilson T, Murray B, Price T, Atherton D, Hooks T.  Non-Nutritive (Artificial) Sweetener Knowledge among University Students. Nutrients. 2019; 11(9). pii: E2201. doi: 10.3390/nu11092201

Thanks for quoting this reference, which was included in the Discussion section. Lines: 315-318

Reviewer 2 Report

This is a very interesting study looking at the impact of food policy on intake of non-nutritive sweeteners in children in Chile. I think this is an important study, and I think the design, analysis, and write-up have all been done well. I have no major concerns, only two minor comments:

I did not realise that this study utilized face-to-face interviewing (i.e., trained nutritionists administering FFQ to participants) until I read lines 346-351 in the discussion. When I originally read the paper, I thought participants completed the FFQ by themselves, and I did wonder about the validity of the younger children’s data. It would be helpful to clarify the methodology used (in Section 2.3, lines 136-142). I see that it currently states “The application of this instrument was done by trained nutritionists,” but I think this could be made clearer.

I think it could be made a bit more explicit that this was an in-person study. The only reason I realised this was an in-person study was because of the section on measuring height and weight (and then when I read “face-to-face” in the discussion). It might just be useful to state where the study took place, “participants attended a x-min session in the lab" for example.

Author Response

RESPONSE TO REVIEWER 2

I did not realise that this study utilized face-to-face interviewing (i.e., trained nutritionists administering FFQ to participants) until I read lines 346-351 in the discussion. When I originally read the paper, I thought participants completed the FFQ by themselves, and I did wonder about the validity of the younger children’s data. It would be helpful to clarify the methodology used (in Section 2.3, lines 136-142). I see that it currently states “The application of this instrument was done by trained nutritionists,” but I think this could be made clearer.

We fixed the text as suggested: “The application of this instrument was done by trained nutritionists through face to face interviews with parents/tutors and children surveyed in this study”.

I think it could be made a bit more explicit that this was an in-person study. The only reason I realised this was an in-person study was because of the section on measuring height and weight (and then when I read “face-to-face” in the discussion). It might just be useful to state where the study took place, “participants attended a x-min session in the lab" for example.

As suggested, the first paragraph of the Methods section was changed to: “An observational cross-sectional study was conducted on 250 children aged 6 to 12 years residing in Santiago, the country capital city located within the Metropolitan Region of Chile, together with their parents or tutors by face-to-face interview and anthropometric evaluation in schools.
